# Implementation of Telehealth Among Older People: A Challenge and Opportunity for Latin America and the Caribbean—A Literature Review

**DOI:** 10.3390/healthcare13212680

**Published:** 2025-10-23

**Authors:** Rafael Pizarro-Mena, Elena S. Rotarou, Hendrik Adrián Baracaldo-Campo, Samuel Duran-Aguero, Solange Parra-Soto, Felipe Retamal-Walter, Patrick Alexander Wachholz, Silvia Maranzano, Victoria Tirro, Sara Aguilar-Navarro, Isabel Barrientos-Calvo, Valeria Carpio-Arias, Clarissa Botello, María Fernanda López, Roni Mukamal, Alessandra Tieppo, Igor Cigarroa, Fausto Medola, Gloria Riveros-Basoalto

**Affiliations:** 1Facultad de Ciencias de la Rehabilitación y Calidad de Vida, Universidad San Sebastián, Sede Los Leones, Santiago 7500000, Chile; samuel.duran@uss.cl; 2Núcleo Milenio Estudios en Discapacidad y Ciudadanía—DISCA (NCS2022_039), Santiago 7500000, Chile; elena.rotarou@uss.cl; 3Facultad de Medicina, Universidad San Sebastián, Sede Los Leones, Santiago 7500000, Chile; 4Facultad de Ciencias de la Salud, Universidad Autónoma de Bucaramanga—UNAB, Floridablanca 680003, Colombia; hbaracaldo@unab.edu.co; 5Departamento de Nutrición y Salud Pública, Facultad Ciencias de la Salud y de los Alimentos, Universidad del Bio-Bio, Chillán 3780000, Chile; sparra@ubiobio.cl; 6School of Health and Rehabilitation Sciences, The University of Queensland, Brisbane QLD 4072, Australia; f.retamalwalter@uq.edu.au; 7Faculdade de Medicina de Botucatu, Universidade Estadual Paulista—UNESP, São Paulo 18618-687, CEP, Brazil; patrick.wachholz@unesp.br; 8Departamento de Educación Física, Universidad Nacional de Luján, Delegación San Fernando, Buenos Aires 164, Argentina; sprivada@unlu.edu.ar; 9Departamento de Ciencias del Comportamiento, Escuela de Psicología, Facultad de Ciencias, Universidad Metropolitana, Estado Miranda 76819, Venezuela; vtirro@unimet.edu.ve; 10Instituto Nacional de Ciencias Médicas y Nutrición Salvador Zubirán, Ciudad de Mexico 14000, Mexico; sgan30@hotmail.com; 11Unidad de Investigación Hospital Nacional de Geriatría y Gerontología, Universidad de Costa Rica, San José 11501-2060, Costa Rica; isacrisba80@gmail.com; 12Grupo de Investigación en Alimentación y Nutrición (GIANH), Escuela Superior Politécnica del Chimborazo, Riobamba 060104, Ecuador; tannia.carpio@espoch.edu.ec; 13Departamento de Medicina, Facultad de Medicina, Universidad Nacional de Panama, Ciudad de Panama 0816-01552, Panama; clarissabotello24@gmail.com; 14Facultad de Psicología y Ciencias Sociales, Universidad Flores, Buenos Aires C1406EEE, Argentina; licmflopez@gmail.com; 15Servicio de Geriatria e Gerontologia, Departamento de Clínica Médica da Faculdade de Medicina, Universidade Federal do Espírito Santo, Vitória 29055-420, ES, Brazil; ronimukamal@yahoo.com.br; 16Universidade Anhembi Morumbi, São Paulo 03164-000, CEP, Brazil; atieppo@uol.com.br; 17Escuela de Kinesiología, Facultad de Ciencias de la Salud, Universidad Católica Silva Henriquez, Santiago 7500000, Chile; icigarroac@ucsh.cl; 18Facultad de Ciencias de la Salud, Universidad Arturo Prat, Victoria 4720000, Chile; 19Departamento de Design, Faculdade de Arquitetura, Artes, Comunicação e Design (FAAC), Universidade Estadual Paulista “Júlio de Mesquita Filho”—UNESP, Bauru 17033-360, CEP, Brazil; fausto.medola@unesp.br; 20Bibliotecóloga Referencista, Vicerrectoría Académica, Universidad San Sebastián, Santiago 7500000, Chile; gloria.riveros@uss.cl

**Keywords:** older people, telehealth, telegerontology, telegeriatrics, Latin America, COVID-19

## Abstract

Although the COVID-19 pandemic negatively affected the health and quality of life of older people (OP), it provided an opportunity for the implementation of telehealth in the areas of Gerontology and Geriatrics, globally and in the countries of Latin America and the Caribbean, which enabled the continuity of healthcare interventions. Therefore, this literature review aimed to (a) conceptualize telehealth in OP through the lens of Gerontology and Geriatrics; (b) analyze the implementation, facilitators, and barriers of telehealth for OP during the COVID-19 pandemic at both global and Latin American and Caribbean regional levels; (c) identify lessons learned and considerations for improving implementation and reducing barriers to telehealth for OP; and (d) discuss challenges related to the integration of telehealth for OP in the region. The databases consulted were PubMed, Scopus, and Scielo; scientific articles in both English and Spanish were considered, including research conducted globally and in Latin American and Caribbean countries that contributed to the objectives of the literature review; the search was conducted from the year 2020 onwards. In addition, government documents and non-governmental technical guidelines from countries in the region were included, whether they focused specifically on older populations or the general population; the search was not limited to a specific time period. Initially, in our search strategy, 1631 scientific articles and 20 governmental and non-governmental documents were identified for the literature review. After eliminating duplicate and applying the inclusion and exclusion criteria, 84 documents were selected for the literature review (46 analyzed the implementation, barriers, and facilitators of telehealth during the COVID-19 pandemic). This literature review presents a conceptual analysis of the implementation and facilitators of, as well as barriers to, telehealth among OP during the COVID-19 pandemic from the perspective of healthcare providers and OP themselves. The paper synthesizes a number of international and Latin American experiences and proposes several recommendations for the implementation of telehealth for OP in the Latin American and Caribbean region. Despite the ongoing challenges regarding telehealth research and training, this review describes telehealth for OP as an intervention approach that improves the provision of holistic care, favoring OP autonomy, functionality, and overall quality of life. This review also proposes telehealth as a regular intervention approach to clinical practice in Gerontology and Geriatrics in the region. Collaborative endeavors are needed to further regulate and promote public policy on telehealth, telemedicine and telerehabilitation for OP.

## 1. Introduction

Worldwide, the population aged 60 and over will increase from 900 million to more than 1.4 billion people between 2015 and 2030. This represents a 64% increase in just 15 years, making it the fastest-growing age group [1]. Although the situation varies significantly across regions, the Latin America and the Caribbean region is on the verge of an unprecedented shift in its history, with projections indicating an increase from 76 million older persons (OP) today to 147 million by 2037 [1]. Although the region as a whole has entered a stage of accelerated aging, in half of the countries (some of which are among the least developed), the process is still incipient and moderate, with the most significant changes expected to occur between now and 2030 [1].

Although the Coronavirus SARS-CoV-2 (COVID-19) pandemic was a global public health emergency between 2020 and 2023, its health consequences remain a priority for the region [2]. Quarantines, isolation measures, and mobility restrictions were public health strategies implemented during the pandemic. Among OP, longer quarantine periods were observed, overlooking their heterogeneity, health and illness history, life trajectories, and the diverse ways in which people age. This approach disregarded their role as agents of change and full-rights citizens [3], shifting from physical distancing to a form of social exclusion. This distancing had a negative impact on the overall health of OP, defined as a geriatric emergency [3] associated with physical inactivity [4,5], increased susceptibility to chronic diseases and geriatric syndromes [3], and related mental health issues [6,7].

Specifically, various bio-psycho-social challenges were identified among OP during the COVID-19 pandemic. For example, low levels of physical activity (PA) were associated with immunosenescence, increased risk of chronic diseases, frailty, cognitive decline, and mortality [8]; a decrease in muscle mass, muscle strength, and the number of steps per day [4]; sleep problems, insomnia, and daytime sleepiness [3,9]; worsening of pre-existing conditions, such as diabetes, hypertension, angina pectoris, cardiac events, and psychiatric conditions [10]. Poor appetite control was also described (including possible excessive consumption of ultra-processed foods, risk of malnutrition, weight gain or loss), along with reduced exposure to sunlight and vitamin D [9]. The loss of daily routines and the reduction in physical and social contact led to a sense of isolation from the rest of the world, boredom, frustration, uncertainty about the future, and fear of infection [11,12], resulting in stress, distress, and agitation. The situation was further aggravated by the inability to carry out Activities of Daily Living (ADLs) [11,12]. This was accompanied by limited participation in community activities, formal physical activity, and adherence to medical regimens, and public health recommendations [13]. Additionally, the discontinuation of participation in cognitive stimulation activities, workshops, social gatherings, group therapies, and/or volunteer work contributed to increased cognitive decline and depressive symptoms, resulting in reduced social contact and heightened feelings of loneliness [3]. A complex and bidirectional relationship was even described between mental health problems and social disconnection [10], leading to the emergence of social frailty among OP [13]. All these bio-psycho-social challenges affected the resilience that OP had developed throughout their aging process prior to the pandemic.

This global health situation also affected the implementation and continuity of government-led programs for OP, due to lack of access [14]. Likewise, it disrupted in-person research with OP, forcing projects to either suspend operations or implement remote, contactless methods [15,16]. In this context, and as alternatives to help OP maintain their ADLs, the World Health Organization (WHO) encouraged continued social engagement through virtual means and regular physical activity at home [17]. In doing so, the pandemic accelerated innovation in telecommunications and normalized the use of previously underutilized services and products, such as telehealth [15,18], understood as, a modality that utilizes Information and Communication Technologies (ICTs) to pro-vide health services, medical care, and information regardless of distance [19]. This led to an earlier-than-expected implementation of telehealth globally, across various contexts, programs, and services, and for different health conditions, chronic diseases, and geriatric syndromes [20,21,22,23,24,25,26,27,28,29,30,31,32]. In turn, telehealth maintained and/or improved health outcomes by increasing access to care for OP with limited resources, those living in remote geographic areas, individuals with cognitive and/or physical disabilities, or with limited access to transportation; it also enhanced follow-up and quality of care, reduced costs, and facilitated the collection of collateral information during the pandemic [20,33,34,35,36,37,38]. Additionally, it led to the establishment of new models of care [39]. In this context, at the beginning of the pandemic, a scoping review describing the implementation of telehealth for OP, showed a greater experience of telehealth in high-income countries in the Northern Hemisphere. It also indicated a lack of consistency in the concepts associated with telehealth for OP [40], that is, for the field of action of Gerontology and Geriatrics.

Despite a growing body of literature describing the implementation of telehealth in the health systems of some Latin American countries, some countries stand out as having pushed more than others to adopt telemedicine in their health systems, as Argentina, Brazil, Chile, Colombia and Mexico [41], as well as some of the main facilitators and barriers to telehealth in the region [42], these studies have not focused specifically on OP. Therefore, this literature review aimed to (a) conceptualize telehealth in OP through the lens of Gerontology and Geriatrics; (b) analyze the implementation, facilitators, and barriers of telehealth for OP during the COVID-19 pandemic at both global and Latin American and Caribbean regional levels; (c) identify lessons learned and considerations for improving implementation and reducing barriers to telehealth for OP; and (d) discuss challenges related to the integration of telehealth for OP in the region.

## 2. Materials and Methods

### Search Strategy

A literature review was chosen as the methodological approach, as it allows for a broader and more diverse identification of the available literature, as well as a more flexible and context-sensitive exploration [43]. To achieve the objectives of this literature review [43], a bibliographic search was conducted to answer the question: How was telehealth implemented for older people during the COVID-19 pandemic at the global level and in Latin America and the Caribbean region? The keywords and search terms (MeSH/DeCS), in both English and Spanish, were as follows: older adult, older person, older people, old age, aging, aged, elderly, telehealth, telemedicine, telerehabilitation, Covid, Pandemic, SARS-CoV-2, Coronavirus, implement, facilitator, and barrier. The search strategy is described in Appendix A, Table A1.

The databases consulted were PubMed, Scopus, and Scielo. Scientific articles in both English and Spanish were considered, including research conducted globally and in Latin American and Caribbean countries that contributed to the objectives of the literature review. The search was conducted from the year 2020 onwards.

In addition, government documents and non-governmental technical guidelines from countries in the region were included, whether they focused specifically on older populations or the general population. To do this, the research team, made up of members from different countries (Mexico, Costa Rica, Panama, Venezuela, Colombia, Brazil, Ecuador, Argentina, and Chile), reviewed the websites of the health ministries, professional associations, and other non-governmental organizations from countries in the region. The documents included addressed the implementation of telehealth in the countries of the region. The search was not limited to a specific time period.

Conference abstracts, scientific conference reports, dissertations, research reports, theses, case studies, protocols, letters to the editor, and editorials were excluded. The inclusion and exclusion criteria are described in Appendix A, Table A2.

The literature review process followed the steps outlined in references [44,45], depending on the type of document. In the case of scientific articles, the steps included were: search strategy, article identification, removal of duplicates, title and abstract screening, full-text review, selection, data compilation into tables, analysis, interpretation of results and discussion, and referencing. For governmental and non-governmental documents, the steps were: search, identification, full-text review focused on the implementation of telehealth for both OP and the general population in countries of the region, selection, data compilation into tables, analysis, interpretation of results and discussion, and referencing. A reference librarian, part of the research team, contributed to the various stages of the literature review process. This professional contributed to the process by helping with defining keywords, formulating the search strategy, identifying articles, removing duplicates, retrieving the full text of articles for review (from the university’s institutional databases), proposing tables for organizing the results, and compiling the bibliography.

Initially, 1631 scientific articles and 20 governmental and non-governmental documents were identified for the literature review. A total of 254 duplicates were found and removed. After applying the inclusion and exclusion criteria, 84 documents were selected for the literature review (76 scientific articles, 4 governmental documents, and 4 non-governmental documents). Figure 1 presents a flowchart of the article selection process for the literature review.

## 3. Results

### 3.1. Conceptualization of Telehealth in Gerontology and Geriatrics

Guidelines from various international, governmental, and non-governmental organizations, both prior to and during the COVID-19 pandemic, they were identified, and thus, established conceptual frameworks related to telehealth [19,46,47,48,49,50]. Consequently, telehealth is understood as a modality that utilizes Information and Communication Technologies (ICTs) to provide health services, medical care, and information regardless of distance [19]. Telehealth is also a tool for disseminating information on the care and prevention of chronic diseases, as well as epidemics [19]. Telehealth is also part of AgeTech, understood as the use of advanced technologies for OP, including both existing devices (such as eyeglasses, walkers, and prosthetics) and emerging technologies (digital media, ICTs, mobile technologies, wearables, and smart home systems) [51], which experienced significant growth during the COVID-19 pandemic [51].

Three key components of telehealth are described: tele-education, telemedicine, and tele-assistance [19,47]. Specifically, telemedicine focuses on the provision of remote health services in the areas of health promotion, prevention, diagnosis, treatment, and rehabilitation. These services are delivered by healthcare professionals who use ICTs to exchange data, with the aim of facilitating access and timely delivery of services for populations facing limited healthcare availability, access barriers, or both, in their geographic area [19,46]. Telerehabilitation, in turn, focuses on the provision of rehabilitation services remotely through the application of telecommunication, remote sensing, operational, and information technologies. Its purpose is to serve individuals, professionals, and systems by minimizing barriers of distance, time, and cost [19,46,50]—a concept that was further reinforced during the COVID-19 pandemic [50]. In summary, telerehabilitation is a branch of telemedicine [50], and telemedicine is a branch of telehealth [19]. Regarding its modality, there are three ways to implement telehealth, telemedicine, and telerehabilitation: synchronous (in real time between the professional and the user); asynchronous (at different times and/or for later review); or hybrid (a combination of both) [19,46].

Telehealth has been incorporated into public health systems in some countries of the region [41]. For example, in the case of Chile, telehealth is understood as a strategy based on the Comprehensive Family and Community Health Care Model, within the context of Integrated Health Service Networks. It facilitates the provision of remote services in the areas of health promotion, prevention, diagnosis, treatment, rehabilitation, and palliative care [19], with a person-centered approach that considers the sociocultural context and life course. The aim is to maintain optimal health status and continuity of care for the population, thereby improving equity in access, rights, timeliness, and quality of care [19]. This intervention was implemented during the pandemic to ensure continuity of care for OP in primary healthcare (PHC) programs [52].

Integrating the conceptualization of telehealth (its components and branches) previously described [19] within the fields of Gerontology (the science of aging) [45,53] and Geriatrics (a branch of this science that focuses on addressing the needs, problems, and geriatric syndromes of OP across different levels of healthcare) [45,54], we propose the concept of Telegerontology, expanding on previous definitions [55]. Additionally, we expand the concept of Telegeriatrics as proposed by the Brazilian Society of Gerontology and Geriatrics [56] for broader application in the region, as summarized in Table 1 and Figure 2.

These updated concepts, which reflect the regional experience of Latin America and the Caribbean, allow for a more precise understanding of the scope of Telehealth (as a care modality) in the fields of Gerontology and Geriatrics, as well as in the daily approach to OP in different contexts, programs, and services; concepts that can also be used in other regions globally (Table 1). Update and integrate these new concepts “Telegerontology” and “Telegeriatrics” into previously published concepts in the field of Telehealth [19], provides a more appropriate representation for the fields of Gerontology and Geriatrics (Figure 2).

### 3.2. Overview of the Implementation, Barriers, and Facilitators of Telehealth for Older People

This literature review has compiled a total of 46 scientific articles that address the implementation, barriers, and facilitators of telehealth for OP (Appendix A, Table A3). Among these, an integrative review article, published prior to the COVID-19 pandemic, has been included to provide an initial comparative perspective on the barriers and facilitators of telehealth for OP.

Several types of research have been identified: Quantitative (20), Qualitative (12) and Mixed Methods (6). Within quantitative research, primarily through cross-sectional studies. Within qualitative research, primarily through semi-structured interviews. A similar trend can be observed in research using mixed methods. This reflects the need and the resources that researchers had available at the beginning of the pandemic to address this research topic, as well as the diversity of information collected. Furthermore, there are studies that have included some type of intervention (10), primarily analyzed at the end of the intervention.

Most studies focused exclusively on OP, healthy individuals and/or those with various chronic diseases (23), followed by groups of physicians and/or healthcare professionals (7), and then both groups combined (6). In the case of studies using exclusively quantitative methods (excluding mixed methods), the sample sizes ranged from 8 to 3257 OP. For studies using exclusively qualitative methods (excluding mixed methods), the sample sizes ranged from 7 to 98 OP. This reflects the diversity of stakeholders who have reported the information. Two studies presented data on the number of consultations with the Telehealth service.

Globally, the United States is the main country of origin for these studies (15), followed by Canada (5). Specifically in Latin America and the Caribbean, eleven articles on this topic have been identified, published in Chile (5), Mexico (2), Brazil (2), and Peru (2).

Eight review articles were included, providing a broader perspective to support the objectives of this literature review. Three of these review articles specifically included information from countries in Latin America and the Caribbean.

Next, and in the following three sections, we present a narrative and detailed analysis of the research findings, with regard to implementation, barriers, and facilitators of telehealth for OP.

### 3.3. Implementation of Telehealth for Older People During COVID-19

Older people are the fastest-growing group of ICT users [15]. A study in 1067 OP in Italy conducted during the pandemic identified that smartphones were the most commonly used devices for internet access (59%), WhatsApp was the most frequently used app (57%), internet users were younger than non-users (63 vs. 78 years old), and they had higher educational levels [57]. Furthermore, a study conducted in Chile among 680 OP who used technology during the pandemic showed that 91% of participants considered having internet access useful in their daily lives, and 86% stated that the internet enabled them to access knowledge they would not have obtained otherwise. In addition, participants reported using social media in their everyday lives, including WhatsApp (95%), Facebook (82%), and YouTube (60%) [58].

Regarding the use of telehealth among OP, it increased from 4.6% before the COVID-19 pandemic to 21.1%, with 1 in 3 OP having used telemedicine during the pandemic [32]. Since the onset of the pandemic, there has been growing interest and acceptance of telehealth among OP, with studies indicating that active collaboration between OP and their healthcare providers has been linked to increased acceptance of remote services [40]. It was also reported that OP were more motivated and able to quickly learn new technologies in the context of the pandemic [59], enabling them to continue with their ADLs, exchange informal support with family and neighbors, rely on formal support from community organizations, and remain physically active as part of their resilience [60].

A scoping review investigating the use of telehealth in geriatrics revealed that telehealth interventions targeting OP at the onset of the pandemic focused predominantly on curative services, rather than on promotional—preventive care and rehabilitation [40]. However, these services were predominantly available in high-income countries of the Northern Hemisphere, compared to those in the Global South and middle- to low-income countries [40], such as those in the Latin American region. In addition, there was a greater focus on research about telehealth for OP based on information provided by healthcare providers, both during the beginning of the pandemic [61,62,63], and prior to it [64].

Several international experiences [13,65,66,67,68,69,70,71] demonstrated that telehealth made it possible to ensure continuity of care for OP and reduce the risk of infection during the pandemic [23,72,73]. However, at the global level, there is still a lack of studies that explore in depth issues such as the safety, usefulness, usability, scalability, cost-effectiveness, facilitators and barriers, and demand for the use of telehealth in gerontology and geriatrics. Therefore, there is a clear need for research in these areas now that the COVID-19 pandemic has ended [40], so that the lessons learned can help identify the specific needs of OP when designing and managing future telehealth interventions [65].

Potential uses of telehealth for OP include: remote monitoring of chronic diseases (e.g., vital signs, weight, blood glucose), receiving laboratory results and medical reports, access to personal medical records, and medical/professional consultations via videoconference [9]. Potential uses of telerehabilitation for OP have also been proposed, including: facilitating transitions of care between different healthcare settings (e.g., acute care hospitals, rehabilitation centers, long-term care facilities, home care), preventing falls and reducing functional decline, cognitive rehabilitation, targeted therapy for specific conditions (e.g., heart disease, osteoarthritis), addressing loneliness and social isolation, and monitoring safety in the home environment [9].

A systematic review exploring the application and implementation of telehealth services for OP during the pandemic identified the following services: triage and monitoring, remote monitoring and treatment, online follow-up for residents in healthcare facilities, and the use of online services involving various types of telehealth applications. These applications included phone calls, live videoconferencing, online programs, smartphone apps, Internet of Things (IoT), cloud computing, and social media—mostly reported in countries in the Northern Hemisphere [66]. To ensure the security and privacy of telehealth services, the U.S. Department of Health and Human Services provided a list of video communication providers that comply with the Health Insurance Portability and Accountability Act (HIPAA) [47].

Regarding the specific analysis of the implementation of telehealth with OP during the pandemic, a qualitative study conducted in 2020 through interviews with geriatricians, primary care physicians, and emergency doctors in both rural and urban settings in the United States described telehealth as a more flexible, value-based, and user-centered mode of care [61]. The authors identified several benefits of using telehealth with OP, including the reduction in delayed care and the increase in timely attention, improved efficiency of physicians, enhanced communication with caregivers and OP, reduced travel burdens for the elderly, and greater facilitation of health education and outreach [61]. The same study described several challenges, including unequal access in rural areas and the care of OP with cognitive impairments [61]. Other reported benefits have included greater time efficiency and the delivery of visual information through videoconferencing, as well as increased convenience and safety [34,74,75].

Moreover, in the context of PHC in the United States, a qualitative study conducted in 2021—including interviews with 29 OP and three focus groups with professionals—found that telemedicine contributed to continuity of care. Also, in this same country, another research mixed-method carried out in 80 OPs identified socio-emotional connection and comfort as benefits, and technological barriers and loss of the face-to-face experience as challenges [76]. It was particularly convenient when there was a pre-existing or established doctor–patient relationship or when addressing minor health concerns. It was also beneficial for individuals with reduced mobility, as it reduced their exposure to potentially high-risk environments [23].

From a telehealth management perspective, a series of strengths, weaknesses, threats, and opportunities related to telehealth for OP were synthesized at the beginning of the COVID-19 pandemic [40]. Identifying them would allow for the design and implementation of more optimal telehealth strategies for OP (in both intervention and/or research contexts), whether during a pandemic or under normal circumstances. This also highlights the need to further explore this analysis both during and after the pandemic.

#### Implementation of Telehealth for Older People, Before and During COVID-19 in Latin America and the Caribbean

In the Americas, in the 20 years prior to the COVID-19 pandemic, there has been an increase in resources and technical capacity, an improvement in digital education, patient empowerment in their treatment, and greater public interest in telemedicine; he formation of interdisciplinary teams, academic and professional networks, and virtual medical consultations, have been considered particularly successful [42]. During the pandemic, Chile, Brazil, Colombia, Mexico, Argentina and Peru were the countries that made the most progress in the field of telehealth in Latin America, both within their health systems and/or through legislation, increasing its use in recent years [41].

Specifically regarding OP, during the pandemic, the Chilean Ministry of Health urged the teams of the primary healthcare program “Más Adultos Mayores Autovalentes (+AMA)” to implement the intervention through telehealth modalities (synchronous, asynchronous, and/or supported by manuals), depending on local resources and training [52]. This approach incorporated local experiences and was seen as an adaptation to alternative forms of contact. Other countries in the region have also developed recommendations for implementing telehealth for OP. For instance, at the end of 2023, the Brazilian Society of Gerontology and Geriatrics published the Telegeriatrics Guide [56]. However, there are still few guidelines or recommendations on telehealth with an emphasis on gerontology and geriatrics in the region.

Additional benefits have been reported regarding the implementation of telehealth with OP from the perspective of healthcare personnel and patients. A user satisfaction survey conducted during the COVID-19 pandemic in Peru reported that, out of a total of 129 healthcare workers, more than 55% of professionals expressed satisfaction with telehealth [77]. On the other hand, approximately 77% of 377 patients (including 44 OP) reported being satisfied with telehealth services. [77].

Specifically, published experiences have been identified on the implementation of telehealth interventions focused on OP in countries of the region, such as the psychosocial teleassistance program of Peru’s Social Security, which provided 154,280 follow-ups and 36,492 teleassistance services to OP—mainly involving emotional support and social counseling [78]; the TeleCOVID-MG program in two municipalities in Brazil, which offered support to OP [79]; a multidisciplinary intervention conducted in Mexico involving 44 OP with colorectal or gastric cancer from a geriatric oncology clinic, where geriatric assessments, treatment toxicity evaluations, physical exams, and treatment prescriptions were performed using platforms such as WhatsApp or Zoom [80]; a semi-presential technological platform (mobile dental clinic combined with a digital platform) designed to support urgent and priority dental care for OP in five regions of Chile [81]; and telehealth-based evaluations, such as the use of the Phototest to identify mild cognitive impairment in OP with memory complaints from rural areas in southern Chile [82]. Additionally, there are reported experiences that have included OP (average age of 68 years), such as a multidisciplinary supportive care program for individuals with advanced cancer in Mexico. In this program, the most common interventions were psychological care (33%), pain and symptom management (25%), and nutritional counseling (13%). Half of the interventions were delivered via videoconference [83]. Also, in Chile, an online course on healthy aging was implemented for the community and health professionals, which included 20% of OP [84].

Additionally, some interventions initially delivered in person were later continued via telehealth. One example is Brazil’s Playful Living Program, aimed at delivering clown-based performances to OP with aphasia [85]. This telehealth-based intervention included not only clowning but also dance, storytelling, and cooking (safe complementary therapies), delivered through individual and group video calls. It involved the participation of professionals and undergraduate and graduate students from the fields of Speech and Language Therapy, Theater, Psychology, Dance, Culinary Arts, Psychiatry, and Public Health [85]. In addition, to facilitate access for OPs, they received donated technological devices; to improve affordability, they were provided with internet credit donations necessary to stay connected; and to support digital capacity, assistance was offered to help them use the technology [85]. Also in Chile, +AMA—the government program that began in 2015, [52], transitioned from an in-person format to an asynchronous telehealth modality, as a pilot in one of the cities where it is implemented. In its in-person format, it is a health promotion and prevention intervention within PHC, which provides physical activity, cognitive stimulation, and education to OP in the community. During the pandemic, the local team transitioned to an asynchronous telehealth modality through videos uploaded to YouTube and infographics stored on Facebook, which were shared weekly via WhatsApp with OP groups so they could carry out the activities [71]. Monthly telephone follow-ups were conducted. Additionally, during the final month of the intervention, a manual containing the same exercises/activities (including a weekly check-up schedule) was delivered in person to the OP, along with other materials. This intervention has been described in detail previously [71].

### 3.4. Telehealth Facilitators for Older People, Before and During COVID-19

Before the COVID-19 pandemic, an integrative review identified several facilitators that contributed to the acceptance of telehealth by OP with chronic physical conditions. These included: devices with fewer buttons, automatic information transmission, use of low-tech platforms (such as telephones or televisions), devices that provided reminders or alerts (either visual or audio-guided), and user-friendly images [64]. In addition, a systematic mixed-methods review analyzing telemedicine in OP in primary healthcare settings (mostly from studies published before the pandemic) identified the most frequently cited facilitator as location and travel time savings [86].

During the pandemic, qualitative studies involving healthcare providers described additional facilitators of telehealth use in OP [34,63,69]. The first study, conducted in 2020, interviewed 13 healthcare providers in the United States (including medical/clinical directors, program managers, clinical nurses/managers, and clinical social work managers) who provided synchronous telehealth via videoconference to medically complex OP. It identified the following facilitators: assessment of patient needs, collection of information from OP, and increased scheduling capacity [63]. The second study, conducted in the same year, interviewed 33 geriatricians and primary care physicians in the United States to explore their experiences with telemedicine and to identify strategies used to maintain continuity of care for OP [69]. This study reported that many physicians successfully bridged the digital divide by assessing OP’s technological readiness in advance, educating them about telehealth privacy and utility, being flexible with telehealth modalities, using the home or available facilities, making accommodations for disabilities, and involving caregivers [69]. The third study, conducted between 2020 and 2021, interviewed 29 geriatric care physicians in Canada and identified the presence of a caregiver and administrative support as key facilitators of telemedicine use with OP [34]. Additionally, a mixed-methods study conducted in 2020 interviewed 30 professionals—geriatricians, geriatric psychiatrists, and geriatric nurses in Canada—regarding the use of telemedicine technologies with frail OP, their families, and caregivers. The study identified several facilitators, including relative advantage, adaptability, tension for change, available resources, access to knowledge, networks and communication, compatibility, knowledge and beliefs, self-efficacy, champions, external agents, execution, and reflection and evaluation [87].

Other features of synchronous telehealth visits via videoconference have been highlighted as additional opportunities for intervention, as they allow professionals to observe the OP’s home environment [75]—something that in-clinic visits rarely offer. For example, gait assessments revealed mobility obstacles in the home that increased the risk of falls and injuries [65]. Additionally, medication reviews allowed providers to observe how medications were stored and administered [65]. Inviting professionals into the OPs home environment also enabled more direct conversations. Those OPs and caregivers who had previously participated in a video visit reported being more likely to accept a video call visit in the future [65]. At the same time, the use of audio and video equipment of adequate quality (including adaptive devices such as pocket speakers) facilitated these virtual assessments [88]. For its part, regarding asynchronous telehealth, it has been shown that the volume of follow-up, response time, and the quality-related variables involved in service delivery are critically important to ensure patient/user satisfaction, programmatic success, and the sustainability of the intervention [20]. Among the methodologies used, the use of WhatsApp has been described as reducing loneliness among OP [89], and more generally, the use of technology (in various forms) is considered a strategy to address loneliness in OP living alone [90].

A mixed-methods study conducted in the United States in 2020, involving 3 patients, 4 caregivers, 19 physicians, 5 medical assistants, and 2 programers, identified customer service support and the existence of protocols to guide patients in the use of telemedicine as key facilitators of video consultations [91]. Specifically regarding OP, a qualitative study conducted in New Zealand in 2021 aimed at identifying facilitators for the use of telehealth in underserved rural populations—through seven focus groups and thirteen individual interviews across four diverse communities—found that the term *telehealth* was not initially understood by many participants and required clarification [92]. Those who had used telehealth reported positive experiences—such as saving time and costs—and were likely to use it again [92]. Facilitators also included trust in telehealth technology (reliable and accessible internet access, and access to digital devices); trust in the user experience through the relationship between the consumer and healthcare provider (such as the availability of user support); and trust in the healthcare system (regarding wait times, communication and coordination, and cost) [92]. Additionally, facilitators included access to culturally appropriate healthcare services, the type of consultation (enabling access to specialized services), and the implementation of telehealth (through a fixed community center or a mobile bus unit, with support available—especially when combined with non-health services such as online banking) [92].

#### Telehealth Facilitators for Older People During COVID-19 in Latin America and the Caribbean

From the perspective of healthcare providers in the Latin American region, evidence on facilitators of telehealth for OP during the pandemic is limited. However, insights can be drawn from the perspective of OP telehealth users, as demonstrated in a qualitative study conducted in Chile in 2020. This study used in-depth semi-structured interviews with 26 community-dwelling self-sufficient OP (14 participated in the telehealth intervention, and 12 served as non-intervention controls) [71]. These OP participated in the government intervention program +AMA [52]. This is a multicomponent intervention (Physical Activity, Cognitive Stimulation, and Education) that, during the pandemic, was delivered through an asynchronous telehealth modality (videos, infographics, and a manual) to ensure continuity of care [52]. The OP had previously participated in the same intervention in its in-person format in earlier years. Following thematic analysis, the OP identified the following as facilitators: participating in the intervention alongside others, prior experience with the in-person modality, good digital literacy, self-motivation, commitment to the program, and the emergence of natural leaders [71]. In addition, the OP identified several positive aspects of this modality, such as the sustainability of the effects between the in-person and telehealth formats, the bio-psycho-social benefits achieved through telehealth, the fact that telehealth helped establish a routine, the guidance and instructions provided by professionals, and the flexibility to carry out the intervention at any time and place, which fostered a sense of autonomy and flexibility [71].

### 3.5. Telehealth Barriers for Older People, Before and During COVID-19

Prior to the pandemic, the aforementioned review [64], also described intrinsic barriers to the use of technology. These included font size, the use of unusual characters (which are difficult to read), soft graphics and poor color contrast, devices with widgets (which pose challenges for fine eye-hand motor coordination), the use of a computer mouse (difficult for individuals with hand arthritis), lack of experience using a smartphone or computer, multiple screen transitions required to complete tasks, multi-layered menu bars, inappropriate smartphone size (either too large or too small), and difficulties handling devices among many OP with reduced grip strength [64]. Delays in responses, lack of feedback, and technical problems can lead to frustration and decrease motivation for OP to continue engaging in self-care activities. Likewise, for OP who are not accustomed to using technology, telehealth can represent a significant cultural shift [64]. Moreover, in the previously mentioned mixed-methods systematic review [86], the most frequently cited barrier was user habits or preferences.

During the COVID-19 pandemic, additional barriers were described. A quantitative study involving surveys with 16 primary care physicians from a U.S.-based program serving 873 community-dwelling OP aimed to identify barriers to synchronous (video-based) telehealth. The study found that over one-third of these OP participated in video-based telehealth visits for the first time between April and June 2020. It also reported that most required assistance from a family member and/or paid caregiver to complete the visit. Among OP who had not used telehealth, 27% were unable to engage via video due to factors such as cognitive or sensory impairments, while 14% lacked access to a caregiver who could assist them with the technology [62]. Similarly, physicians were unaware of their OP patients’ internet connectivity, ability to afford mobile phone plans, or access to video-capable devices [62]. In another qualitative study conducted in 2020, involving interviews with 13 synchronous (video-based) telehealth providers in the United States who worked with medically complex OP, the reported barriers included cognitive and sensory abilities, access to technology, reliance on caregivers and aides, addressing sensitive topics, and incomplete examinations [63]. A mixed-methods study conducted in 2020 in Canada, which interviewed 30 professionals (geriatricians, geriatric psychiatrists, and geriatric nurses), identified several barriers to telemedicine for frail OP, their families, and caregivers. These included complexity, quality of design and presentation, patient needs and resources, readiness for implementation, and cultural factors [87]. More recently, a qualitative study conducted between 2020 and 2021 with 29 geriatric care physicians in Canada identified several barriers to telemedicine, including difficulties using technology, patients’ sensory impairments, lack of hospital support, and a high volume of pre-existing patients [34].

In the previously mentioned 2020 study conducted in the United States with 33 participants (including 3 patients, caregivers, physicians, medical assistants, and programers), the identified barriers to video consultations included difficulties navigating technology, concerns about privacy, and lack of technical support [91]. Additionally, a quantitative study conducted in 2021 in Canada, involving surveys with 39 healthcare providers, 40 patients with multiple comorbidities, and 22 caregivers from outpatient clinics at a tertiary hospital—who had received phone visits—expressed interest in future telehealth visits. They identified similar barriers: lack of access to technology and skills, the perception that telehealth may be inferior to in-person visits, lack of administrative support, limited technological skills among both healthcare professionals and patients, and limited access to infrastructure and internet [93].

Also, in a qualitative study conducted in 2021 in the United States involving 29 OP from primary care, participants expressed concerns about the lack of eye contact, which they felt led healthcare professionals to overlook essential details. They also identified issues related to poor communication due to language or hearing barriers [23]. Similarly, in a qualitative study conducted in 2021 with 20 OP in Canada, identified barriers to telehealth included the need for physical examination and touch, lack of nonverbal communication, difficulties using technology, and systemic barriers to access [74]. Furthermore, in a qualitative study conducted with 14 OP in Israel, they expressed concern about the potential for poorer quality of telemedicine sessions compared to in-person sessions, and they also expressed a lack of trust in telemedicine services and frustration with the lack of face-to-face interaction and communication [94]. Additionally, in a mixed-methods study conducted in 2021 in the United States with 249 OP residing in independent living facilities in the community, the main barriers to video visits included not knowing how to connect to the platform, lack of familiarity with technology and the internet, difficulty hearing, limited English proficiency, and lack of interest in seeing providers outside the clinic setting [95]. In addition to the above, in a qualitative study of 78 OP in Singapore, they preferred in-person consultations due to a perceived lack of human interaction and diagnostic accuracy, low digital literacy, and a lack of access to telehealth-compatible devices [96].

#### Telehealth Barriers for Older People During COVID-19 in Latin America and the Caribbean

In the Latin American region, while there is limited evidence on telehealth barriers for OP during the pandemic from the perspective of providers, there is information from the perspective of OP users. A quantitative study, which surveyed 237 OP in Brazil participating in the TeleCOVID-MG program, identified certain personal barriers such as being over 80 years old, having lower educational levels, and experiencing hearing and visual impairments—all of which were associated with the need for support during teleconsultations. This need for support was also associated with hospitalization and mortality [79]. Also, in a quantitative study conducted in 2020 in Chile surveying 34 OP, participants received physical exercise and pain education videos over a 4-week period to be performed independently at home (asynchronous telehealth) [97]. The average adherence was 2 days per week over 2.5 weeks, with the main barrier identified as a lack of willingness to perform the activities. This reluctance was more common among individuals who had engaged in group physical exercise before the pandemic [97]. Additionally, in the previously described 2020 study [71], OP who participated in the asynchronous telehealth intervention reported pain during physical activity and the complexity of the cognitive exercises as barriers [71]. Additionally, those who did not participate in the intervention (controls) reported low digital literacy and lack of support from others as barriers. They also cited lack of motivation, lack of time, low digital literacy, limited knowledge of how to use social media, and poor internet connection, as reasons for not engaging in telehealth [71].

Recognizing the various benefits, facilitators, and barriers that telehealth presented for OP during the COVID-19 pandemic—summarized in Table 2—will be essential when designing and implementing future interventions.

## 4. Discussion

### 4.1. Lessons Learned and Considerations to Improve the Implementation of Telehealth and Reduce Barriers for Older People

Recent studies have identified a trend and preference among OP and professionals for using telehealth in combination with in-person interventions, rather than relying solely on telehealth [23,71,74,76,94,96]. This includes starting with initial in-person appointments, followed by telehealth and/or additional in-person visits [75].

In general, the three principles for delivering telehealth to OP proposed by The Collaborative for Telehealth and Aging (C4TA) from the United States—a group composed of providers, geriatric experts, telehealth specialists, and advocates—can be followed [98]. First, care must be person-centered; therefore, telehealth programs should be designed to meet the needs and preferences of OP by considering their goals, family and caregivers, language characteristics, and their readiness and ability to use technology [98]. Second, care must be equitable and accessible; thus, telehealth programs should address both individual and systemic barriers to care for OP by taking into account issues of equity and access [98]. Third, care must be integrated and coordinated across systems and individuals [98]. In addition, we present more specific considerations below to reduce telehealth barriers for OP. These recommendations are presented for application in various countries worldwide, including those in Latin America and the Caribbean, based on the findings identified in this literature review.

#### 4.1.1. Intrinsic and Extrinsic Factors

A review of OP’s perceptions of technology showed that a one-size-fits-all approach does not work, and that technologies must be adapted to individual needs [99]. It should also not be assumed that all OP face difficulties accessing virtual care. [88]. Therefore, it is important to identify OP’s perceptions of ICTs in order to encourage their use, considering intrinsic factors, such as control-oriented attitudes, independence, and the need or requirement for safety; and extrinsic factors, such as usability, feedback, and costs [99].

Active listening should be encouraged, as well as discussing both the advantages and disadvantages of telehealth, considering existing recommendations in these areas [23]. It is important to emphasize the positive benefits that promote autonomy [99], adapt to OP’s lifestyles and expectations, and offer personalized interventions that take into account individual preferences and capabilities [5]. Also, the option to choose between a phone or videoconference assessment should be offered [88], all with the aim of preventing rejection or dropout. Therefore, technologies must be perceived as reliable and effective, if long-term use is to be achieved [99].

#### 4.1.2. Digital Health Literacy

It has been reported that older age, low educational level, low income, depressive symptoms, low cognitive functioning, disability, limited support networks, complex application interfaces, and lack of digital health literacy are negatively associated with the use of telehealth [32,57,100,101,102,103,104]. It is important to note that when introducing new ICTs, the steep learning curve that OP may experience should be acknowledged, and support should be provided to help them become familiar with these technologies [5]. In addition, they may feel less comfortable, lack access to the necessary technology, and/or have sensory difficulties (such as impaired hearing or vision), which can hinder their use of telehealth [65]. Likewise, the “digital divide” resulting from socioeconomic status, age, geographic location, and cultural factors should also be taken into account [5].

In terms of Assessment, this could be achieved through the use of in-person or online surveys, as previously reported [73], to identify OP’s readiness for telehealth. This includes internet use, access to internet-connected devices, experiences and concerns related to this mode of intervention, and perceived barriers—ultimately enabling the development of training programs [73].

In terms of Training, a commonly expressed need among OP has been for professionals to provide educational sessions, workshops, and tutorials on how to use remote care [94]. In this regard, a meta-analysis—where the majority of studies included were from the United States—which aimed at evaluating the effectiveness of digital health literacy interventions for OP, found that these interventions led to significant improvements in OP’s knowledge and self-efficacy [105]. Additionally, it has been reported that in-person teaching, guided by a conceptual framework and delivered over a minimum of four weeks, had a more significant effect [105]. This approach also helps to strengthen relationships and bonds, which later foster trust when using telehealth [88].

In addition, programs aimed at increasing digital health literacy and competencies should include physicians, professionals, and OP, as this is essential to ensure that all parties have the knowledge, confidence, and ability to equitably benefit from emerging innovations, such as telehealth [33]. To this end, experiences such as the one implemented in Chile—where an online course on healthy aging was offered to both the community and healthcare professionals [84]—can be followed, as well as other international initiatives, such as volunteer programs involving university students that increased access to telehealth services for OP during the pandemic [106]. Likewise, training and capacity building for the personnel who will deliver telehealth should be encouraged, provided by clinical professionals with experience in telehealth, along with updates on new digital assessment tools [75].

In terms of Intervention, it is recommended that professionals use a coaching model for OP, which increases OP’s sense of responsibility in each session and enhances relationship-centered communication skills [9]. At the same time, review techniques should be used to ensure that OP adequately understand instructions and training. It is also important to listen to both the content and the emotions expressed by OP [9].

In addition, a series of actions should be considered with the OP, their family member and/or caregiver before a telehealth appointment, that include: explaining the format and purpose of the visit, practicing with providers beforehand, reminding the OP and their support network when and how the call will take place, explaining the need for future in-person consultations, obtaining photo documentation and/or encouraging family participation—all with the aim of ensuring better preparation for subsequent telehealth sessions [75]. This presents a challenge for OP, as it increases their responsibility to report on their health status and to engage in self-examination [9]. During the telehealth appointment, it is recommended to ask the OP to repeat the information and review it again at the end of the session or to write it down, provide opportunities to ask questions during the visit, look at the camera and be more expressive to ensure the message is understood—including careful placement of the microphone to avoid blocking the mouth. If multiple people are present, it is helpful to practice participation and decide in advance who will lead the conversation. All of this aims to improve communication [75].

#### 4.1.3. Support, Adaptations, and Complements

OP should be supported in navigating the telehealth platform as needed, and assistance should be provided to promote self-efficacy and self-management. [23]. Social support and the involvement of a caregiver or family member during telehealth increase the likelihood of use [107] and facilitate the use of technology (e.g., Zoom). During the assessment, their participation can also be helpful for obtaining collateral information and may aid in remembering appointments [88], which becomes even more relevant for OP with disabilities [103].

A commonly expressed need among OP has been to simplify telehealth infrastructure [94]. It has also been reported that when enablers are present (having a cellphone/PC/tablet, knowing how to use them, and having internet access), these factors cease to be barriers [32]. Therefore, regarding telehealth and data use, it is recommended to use lower bandwidth if the platform allows it, provide a tablet and data loan service, use high-quality headsets/microphones that do not cover the mouth [75], use adequately functioning audio and video equipment (including adaptive devices such as pocket speakers) [88], and include a chat feature during the video call [23], all of which can facilitate telehealth.

Specifically for OP in rural areas or with disabilities, telehealth interventions can be adapted with a health promotion focus, drawing on international experiences in which context was adjusted (shorter, smaller, and longer-duration activities) and equity was addressed (e.g., phone formats, automatically generated subtitles) without altering the content [108]. Specifically, and based on the authors’ experience, we present a series of recommendations to consider when implementing telehealth with OP with disabilities, in order to reduce barriers and, as a result, improve access, use, and satisfaction with telehealth in this population group (Table 3). These recommendations can help make telehealth equitable and accessible, as well as integrated and coordinated across systems and individuals [98].

It has even been suggested that in cases where telehealth cannot be implemented at all, support for OP should be provided via telephone or social media. Periodic home visits may also be carried out, delivering infographics, guidelines, and/or manuals for use and later feedback [52], following the example of rural schools where due to geographic inaccessibility, this is a common practice to facilitate interventions. In this context, it was found that using a manual as the sole intervention tool is less effective compared to videos or a combination of both (video plus manual) [71], due to a social presence effect.

A strategy that could improve manual delivery and follow-up/support was drawn from a telehealth study during the pandemic, where the manual included a calendar with scheduled dates that OP were instructed to mark [13]. In this way, the number of sessions completed was self-reported during biweekly phone calls, incorporating motivational, psychological, and practical elements, along with goal-setting and an emphasis on independence—allowing OP to choose the physical activity of their preference. This approach enhanced adherence and empowerment [13], enabling positive relationships between users and professionals, addressing questions, monitoring health and intervention progress, and fostering self-efficacy in OP.

#### 4.1.4. Conceptual Framework for the Implementation and Presentation of Research Related to Telehealth for Older People

Implementation science encompasses a broad range of applicable methodologies to improve the dissemination, implementation, and scalability of effective interventions in behavioral, clinical, healthcare, public health, global health, and educational settings, with the aim of improving the quality of care and health outcomes, while taking into account local realities, and ensuring sustainable implementation at scale [109]. This could be an interesting conceptual framework for the implementation of telehealth in different population groups, and specifically among OP. Unlike most publication guidelines, which apply to a specific research methodology, the Standard for Reporting Implementation Studies (StaRI) and its accompanying checklist cover the wide range of study designs used in implementation research [110]; therefore, this checklist can guide and facilitate the reporting of interventions and research related to the implementation of telehealth for OP. Furthermore, when designing and implementing these interventions, researchers should consider the barriers and facilitators identified in this literature review. The need for a comprehensive description of the context, implementation strategies, and interventions, as well as the presentation of various effectiveness, process, and cost-effectiveness outcomes, poses a challenge for journals with strict word limits for research articles and may require innovative solutions and the use of online Supplementary Materials [110], as well as for researchers themselves.

### 4.2. Challenges for Telehealth for Older People in Latin America and the Caribbean

#### 4.2.1. Considerations from a Perspective of Future Pandemics

It is essential that responses to this past crisis (and to similar ones in the future) identify and prioritize OP, who may be at particular risk of being left behind or excluded during the pandemic response and subsequent recovery phases [14]. This is especially important in less developed countries, which are less likely to have access to ICTs and where needs are even greater [14], as may be the case in countries in the region.

As a result, it is necessary to ensure that community-based services and support for OP are maintained despite future physical distancing measures [14]. This presents a challenge for professional teams in social and health programs—at the institutional level (hospitals, long-term care facilities), in primary healthcare, and in the community—who have traditionally worked in face-to-face modalities, and who faced both the need and the opportunity to shift to telehealth in order to ensure continuity of care during the pandemic. In this sense, telehealth for OP should become a regular component of clinical practice for professionals in Gerontology and Geriatrics. Therefore, professional teams should always design and implement their in-person interventions with the awareness that, in situations like a pandemic, they may need to rapidly transition—within days—to a telehealth model to continue providing care.

It has been reported that, during quarantine periods, preventive telehealth strategies focused on cognitive, behavioral, social, positive, and brief interventions—delivered either online or in person—can improve mental well-being, social affiliation, and support, and reduce perceived loneliness [6]. On the other hand, cognitive skills and social support networks can help OP foster meaningful connection and a sense of belonging, improving social connection and promoting healthy relationships with oneself and others [6]. The minimum components that an intervention should include to protect the psychological and mental health of OP during a pandemic are: developing problem- and emotion-focused coping strategies; strengthening social skills and family relationships; and engaging in regular physical activity [111]. In addition, religion and spirituality have been strongly associated with positive psychological well-being, as they provide support, a sense of belonging, life purpose, and helping alleviate distress in the face of adversity [13]; this will be interesting to investigate and include as part of Telesalud’s interventions with OP of the Latin American and Caribbean region, where there is a high Christian and Catholic proportion.

#### 4.2.2. Considerations from a Management Perspective: Design, Implementation, and Evaluation

In addition, investment is needed in care and support services to ensure that they are adapted to the individual needs of OP, advancing in the design, implementation, and management of cost-effective telehealth interventions that prioritize low cost and take into account the diversity and specific needs of OP [15]. These services should be designed to be accessible and aligned with the changes associated with normal aging, considering and differentiating by sex, age, educational level, functional status, cognitive status, digital literacy, context, among other factors. This approach promotes well-being, helps maintain autonomy and independence [14], and improves quality of life. Furthermore, interventions should include OP and their families in the design, user testing, and standardization of telehealth services.

At the same time, it is necessary to support OP and their caregivers so they can access digital communication or alternative ways to stay in contact with their families and social networks, when physical movement is restricted [14]. For this reason, it is also recommended to include OP and their caregivers in learning programs, as well as to regularly improve their access to ICTs [14]. It is both a challenge and a responsibility for professionals to conduct digital health literacy workshops, thereby fostering better performance of the telehealth strategies implemented.

It would also be essential to design, translate, adapt, and/or validate tools that assess motivation, competence, usability, and/or user satisfaction with mobile devices and telehealth, given that very few tools currently exist [112] and there are still challenges and opportunities in identifying valid tools for the OP population [113], —especially in Spanish or Portuguese in the region—following some early-stage experiences in certain countries [77,97]. These tools could complement those used in the Comprehensive Gerontological Assessment (CGA) [45,114], which is the cornerstone of evaluation in this field, thereby facilitating professional training. Furthermore, there is an opportunity to digitize many scales, questionnaires, or clinical tests routinely used in CGA—either through apps or video consultations—as has already been done with cognitive assessment tools in Chile and other countries [38,82,115,116]. Questionnaires or written materials may even be provided to OP before virtual visits to improve the quality of the evaluation [88]. This entire set of tools could be used to assess telehealth in OP and their caregivers, as part of the CGA, as previously recommended [45].

#### 4.2.3. Considerations from a Research Perspective

In parallel, it is necessary to advance qualitative, quantitative, and mixed-methods research with an interdisciplinary focus involving OP, recognizing the specificities and strengths of the region. This will help explore in greater depth how telehealth was implemented during the COVID-19 pandemic from the perspectives of physicians, professionals (in their diversity), and OP themselves, using methodological strategies already employed—such as online surveys, due to their feasibility and reach [117]. It will also be highly relevant to investigate the benefits, facilitators, and barriers—such as those documented in this article (Table 2)—as experienced by both OP and healthcare providers, as well as the effects of different telehealth modalities (synchronous, asynchronous, or hybrid) on the biopsychosocial health and quality of life of OP, considering previous research, which showed that older people were more likely to recover from negative emotions, improve their well-being, and enhance their quality of life during the COVID-19 pandemic through telehealth services [118].

#### 4.2.4. Considerations from an Academic Training Perspective

The fact that healthcare providers themselves may possess greater knowledge about managing telehealth with OP—following a gerontological and geriatric care model, alongside other intervention modalities—will enable them to develop more appropriate interventions with OP, their families, and caregivers, from a more ethical perspective and with a focus on promoting their rights. Likewise, this new knowledge needs to be incorporated into the training and curriculum of undergraduate and postgraduate students/professionals, as well as into continuing education programs for professionals who work or will work in the health and psychosocial fields with OP.

#### 4.2.5. Considerations from a Healthcare System and Public Policy Perspective

Finally, some countries of the region already incorporate the telehealth as part of the strategies from their health ministries, such as Argentina, Brazil, Chile and Colombia [41]. Also telehealth guidelines in OP emanating from non-governmental organizations such as the Brazilian Society of Gerontology and Geriatrics [56], which can be replicated in other geriatric societies or professional associations in the region, with a focus on OP. These and other countries in the region must demonstrate the viability, scalability, profitability, availability and sustainability throughout the health system, of the telehealth with emphasis on gerontology and geriatrics. Therefore, having greater knowledge of telehealth in gerontology and geriatrics in Latin America and the Caribbean region will generate lessons learned and experiences that can be used to improve public policies, regulations, and health systems in countries where implementation barriers still persist [41], in light of other international experiences [119].

One of the limitations of our literature review is that the methodological quality of the scientific articles was not assessed, given the diversity of designs included in this literature review. However, its strengths include an updated concept of TeleGerontology and TeleGeriatrics; a broad and exhaustive review of the global and regional literature on the implementation, barriers, and facilitators of telehealth for OP during the COVID-19 pandemic; lessons learned and considerations for improving telehealth implementation and reducing barriers for OP, both globally and regionally; and the challenges for telehealth for OP in Latin America and the Caribbean (as a present and future opportunity) in aspects related to future pandemics, management (design, implementation, and evaluation), research, academic training, and the healthcare system and public policies.

## 5. Conclusions

During the COVID-19 pandemic, the effectiveness of telehealth was demonstrated, as it provided continuity and access to gerontological and geriatric care services, while reducing the risk of infection among OP by minimizing the need to visit healthcare facilities. Telehealth, thus, became a new modality of intervention with OP and their support networks, accelerating its adoption by several decades.

As a result, the pandemic was an opportunity to evaluate the current systems for delivering social and healthcare services, both for the general population and for OP. This highlights the importance of assessing, designing, and implementing strategies in a telehealth modality, considering all levels of healthcare, in order to ensure the continuity of regular interventions.

The increase in population aging is one of the major challenges in the coming decades, and one in which the needs and requirements among demographic cohorts will vary considerably, in light of individuals’ life course. In this context, the documented benefits of telehealth support its role as a fundamental and routine component of gerontological and geriatric care, particularly for reaching OP who face barriers to accessing healthcare services, including those in rural and remote areas. Similarly, telehealth is an excellent alternative for ensuring continuity of specialized care in situations involving physical isolation (e.g., pandemics, natural disasters, disability, and critical care), which restrict the social and healthcare participation of those affected. In fact, during the early months of the pandemic, telehealth served as a care alternative for OP with various health conditions. A proactive approach to identifying and addressing telehealth barriers can help maximize virtual interactions for the older population and mitigate healthcare inequities.

In the future, it will be necessary to gather local experiences accumulated during the pandemic, as well as to advance research, university education, professional training, and the development of guidelines, regulations, and public policies on telehealth in the fields of gerontology and geriatrics, reflecting the specific needs and context of the region.

Finally, the recommendations emerging from this review are highly relevant for healthcare professionals who work directly with older people, as well as for decision-makers, in guiding effective intervention strategies for the implementation of healthcare services for individuals with limited access to specialized care. Telehealth should help reduce fragmentation, improve data sharing, enhance communication among stakeholders, and address the sustainability of interventions.

## Figures and Tables

**Figure 1 healthcare-13-02680-f001:**
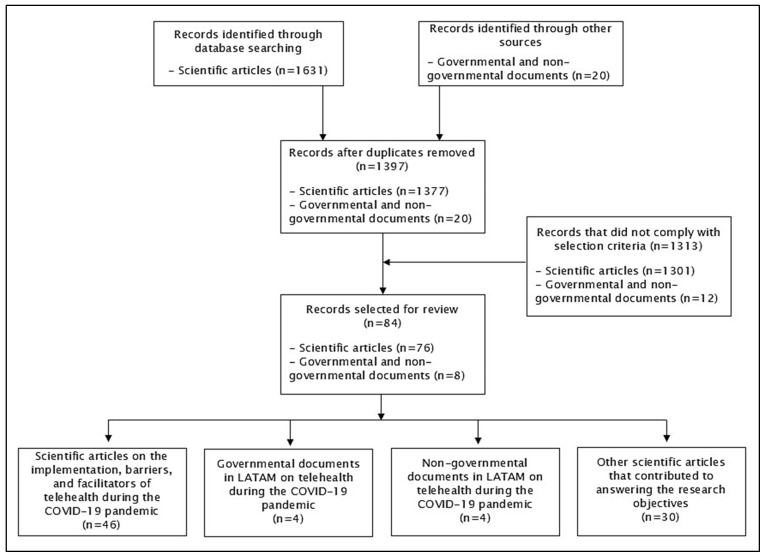
Flowchart of the article selection process for the literature review, classified by type of document. Abbreviation: LATAM: Latin America.

**Figure 2 healthcare-13-02680-f002:**
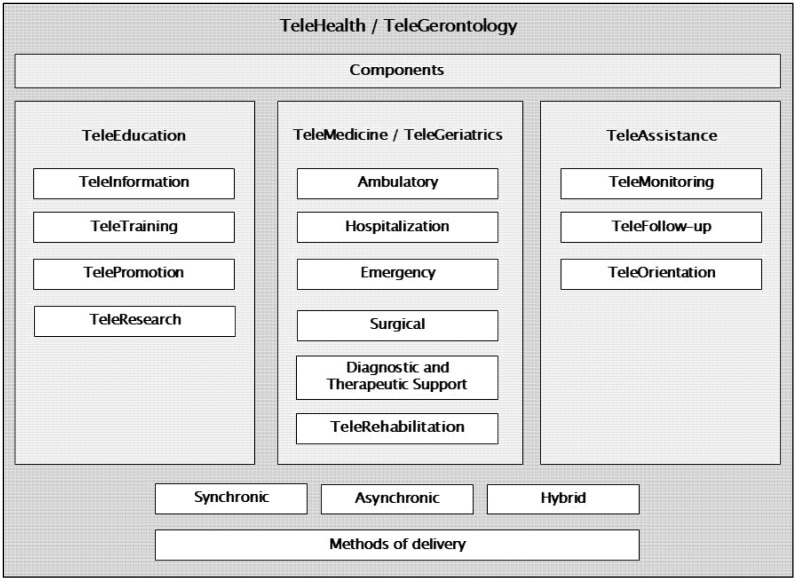
Components of telehealth, their relationship with the branches of telemedicine and telerehabilitation, and their modalities of delivery; with emphasis on the care of older persons, incorporating the concepts of TeleGerontology and TeleGeriatrics. Own elaboration based on the document “National Telehealth Program in the context of Integrated Health Services Networks” [19].

**Table 1 healthcare-13-02680-t001:** Concepts of TeleGerontology and TeleGeriatrics for use in the Latin America and Caribbean region.

Concept	Definition
TeleGerontology	Provision of gerontology services at a distance, through the application of telecommunication, information technology, and computerized data transmission, to support aging and old age across biological, functional, psycho-affective, social, and environmental dimensions of older people and their surroundings; while also understanding what technologies older persons use and how they use information and communication technologies.
TeleGeriatrics	Use of telemedicine to provide older people with access to specialists who can deliver medical services and specialized care to this population [56], with emphasis on health promotion, prevention, assessment, diagnosis, treatment, rehabilitation, palliative care, and management of health problems, chronic conditions, and geriatric syndromes affecting them and their families.

**Table 2 healthcare-13-02680-t002:** Summary of telehealth benefits, facilitators, and barriers among older people during the COVID-19 pandemic: Global and Latin America and Caribbean perspectives.

Context	Benefits	Facilitators	Barriers
Technological/Telehealth	- Provided visual information through video conferences. - Offered a more flexible, value-based, and user-centered mode of care.	- Adaptability of telehealth. - Flexibility with the types of telehealth modalities used. - Using the home or available facilities. - Existence of protocols to guide OP in using telehealth. - Promotion of self-efficacy and self-management among OP. - OP’s trust in telehealth technology (access to reliable and affordable internet, and to digital devices). - Education of OP about the privacy and usefulness of telehealth.	- Telehealth complexity. - Quality of design and presentation. - Limited knowledge of how to use social media. - OP’s difficulty navigating technology. - Limited access to infrastructure and internet. - Poor internet connection. - Lack of eye contact, which can lead professionals to overlook essential details. - Need for physical examination and touch.
Organizational/Governance	- Contribution to continuity of care. - Provision of greater convenience and safety. - Reduction in delayed care. - Increase timely access to care. - Improvement of time efficiency of interventions. - Improvement of efficiency of physicians/professionals.	- OP’s trust in healthcare system (including wait times, communication and coordination, and cost). - Scheduling capacity for care. - Making adaptations for disabilities. - Presence of a caregiver and/or family member. - Type of consultation (allowing access to specialized services). - Videoconferencing facilitating home assessments for OP (environment, medication management). - Follow-up volume, response time, and quality-related variables involved in service delivery. - Provision of customer service. - Having previously participated in the same intervention in person. - Administrative support. - Access to culturally appropriate healthcare services.	- Concerns about privacy and lack of technical support. - Physicians’/professionals’ lack of awareness regarding their OP patients’ internet connectivity, ability to afford mobile phone plans, or access to video-capable devices. - Professionals’ difficulties navigating technology. - High volume of pre-existing patients. - Lack of support in the hospital setting.
Human/Psychosocial	- For OP with reduced mobility, reduction in exposure to potentially high-risk environments. - Improvement in communication with caregivers and OP. - Facilitation of health education.	- OP’s trust based on consumer–health provider relationship (ease of access to user support). - OP’s needs assessment. - Gathering information from OP. - Assessment of technological readiness in advance. - More direct conversations through videoconferencing. - Good digital literacy among OP. - OP’s motivation to use telehealth. - Natural and/or community leaders who support other OP in using telehealth.	- Being over 80 years old. - Lower educational level among OP. - OP’s needs and resources. - Dependence on a family member to complete the videoconference. - Presence of cognitive or sensory impairments in OP. - Lack of communication due to language or hearing barriers. - Addressing sensitive topics and incomplete examinations. - Lack of motivation, willingness, and/or time among OP to engage in telehealth. - No or low digital literacy among OP. - Pain during physical activity and the complexity of cognitive stimulation exercises. - Perception that telehealth may be inferior to in-person visits. - Lack of interest among OP in seeing providers outside the clinic. - Not speaking the native language. - Cultural factors.
Financial	- Reduced the travel burden for OP.	- Time and cost savings. - Flexibility to carry out the intervention anytime and anywhere (asynchronous telehealth).	- Requiring a paid caregiver to complete the videoconference.

**Table 3 healthcare-13-02680-t003:** Recommendations and strategies for implementing telehealth for older people with disabilities, aimed at reducing telehealth barriers.

Type of Disability of Older People	Strategies to Reduce Telehealth Barriers
Visual disability	- Use of screen readers compatible with telehealth platforms. - Adaptation of platforms with voice commands and audio navigation. - High contrast and enlarged font in interfaces. - Usage instructions in audio format. - Training for caregivers on providing remote assistance to OP with low vision or blindness. - Preliminary ophthalmologic evaluation to personalize accessibility. - Implementation of virtual assistants with spoken responses. - Availability of educational materials in braille or audio. - Training in the use of accessible mobile technologies. - Periodic usability assessments of accessible platforms.
Hearing disability	- Inclusion of sign language interpreters in sessions. - High-quality real-time automatic subtitles. - Use of platforms with live chat as an alternative to videoconferencing. - Training for professionals in clear written communication. - Availability of informational materials in sign language. - Visual alerts for notifications and reminders. - Promotion of asynchronous telehealth (email, secure messaging). - Hearing evaluation prior to determining the modality of remote care. - Technological adaptation guides for OP with hearing loss. - Incorporation of sound amplification technologies.
Motor disability	- Interfaces compatible with adaptive devices (trackballs, switches). - Design of simplified navigation platforms (few clicks). - Automation of voice command functions. - Keyboard/mouse configuration adjustments to facilitate use. - In-home training on the use of adapted tablets. - Remote support for personalized equipment setup. - In-home technical assistance when necessary. - Inclusion of caregivers or family members in the telehealth experience. - Assessment of home’s physical environment to ensure accessibility. - Use of ergonomic physical supports to hold devices.
Intellectual disability	- Simple interface design with icons and guided steps. - Use of visual and auditory reminders to schedule sessions. - Digital skills training with the support of a family member and/or caregiver. - Structured repetition of key instructions during the session. - Educational materials adapted in plain language. - Inclusion of caregivers and/or family members in telehealth sessions. - Neuropsychological assessment to tailor technology use. - Shortening session duration based on attention span. - Promotion of digital routines with step-by-step tasks. - Support through digital mentoring (telecompanionship).
Psychosocial disability	- Inclusion of emotional and psychological support before and after sessions. - Use of platforms that ensure confidentiality and privacy. - Reduction in the number of different professionals involved in remote telehealth care. - Flexible scheduling to avoid stressful situations. - Empathetic language adapted to the emotional state of the person. - Inclusion of support networks (family members, caregivers, peers). - Assessment of emotional readiness to use technology. - Digital skills training programs with psychosocial support. - Feedback spaces to express fears and barriers. - Access to mental health resources through integrated platforms.

## Data Availability

No new data were created or analyzed in this study. Data sharing is not applicable to this article.

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
