# Peer review of "Implementation of Telehealth Among Older People: A Challenge and Opportunity for Latin America and the Caribbean—A Literature Review"

_healthcare, 2025, doi:10.3390/healthcare13212680_

Round 1

Reviewer 1 Report

Comments and Suggestions for Authors

1. Lack of Originality and Analytical Depth

Despite the massive volume (over 20 pages) and references to over 80 documents, the manuscript functions more as a descriptive catalog than a critical synthesis of the literature.

- The manuscript predominantly summarizes findings from other papers without advancing any novel conceptual model, hypothesis, or critical framework specific to Latin America and the Caribbean (LAC).

- For example, the supposed contributions such as the definitions of TeleGerontology and TeleGeriatrics (Table 1, Figure 2) are merely repackaged summaries of known WHO definitions, lacking any empirical or conceptual innovation.

- The “lessons learned” in the discussion section (e.g., section 4.1.1–4.2.3) offer generic advice, often pulled directly from pre-pandemic sources or guidelines from high-income countries, not LAC-specific insights.

Conclusion: This fails to meet the criteria for a publishable literature review in a peer-reviewed international journal that requires critical synthesis, theory-building, or policy-relevant insight.

2. Methodological Deficiencies and Selection Bias

The methodology presented in section 2 is insufficiently rigorous for a scholarly review:

- No PRISMA flow diagram or transparent inclusion/exclusion rationale for the final 84 documents.

- The search strategy is vague (“no time limit”), lacks specificity in Boolean logic, and merges governmental reports with peer-reviewed literature without quality assessment.

- There is no critical appraisal or grading of the evidence. The authors treat peer-reviewed RCTs and anecdotal local government reports with equal weight (e.g., \[52] Chile +AMA intervention vs. \[90] pilot video trials).

- The review is regionally biased toward Chile, Brazil, and Mexico, without transparency on why many Caribbean nations or lower-income LAC countries are underrepresented.

Conclusion: The methodological approach does not meet the standards of a systematic or scoping review and invites questions about selection bias and credibility of synthesis.

3. Overreliance on Low-Quality or Redundant Sources

A substantial portion of the cited material (e.g., \[13]-5times, \[52]-6times, \[68]-9times) is repetitively used, often drawn from the same national government programs (especially Chile’s +AMA). This creates an illusion of breadth, while in reality, the review heavily depends on a narrow subset of interventions and does not generalize.

- There is no effort to rank or weigh sources based on study design, size, or outcomes.

- Many of the cited studies are small qualitative or observational case studies, yet conclusions are drawn as if they were generalizable across LAC.

Conclusion: This undermines the review’s validity and generalizability, particularly for policy or regional guideline development.

4. Inflated Claims and Unsubstantiated Recommendations

The authors make unwarranted normative statements without support from the literature.

- Statements such as “telehealth should become a regular component of clinical practice in Gerontology and Geriatrics in LAC” (line 678) are policy-level claims made without demonstrating feasibility, cost-effectiveness, or systemic readiness.

- The paper suggests “telehealth improves quality of life” (abstract) despite citing no meta-analytic or controlled trial data to support such a sweeping claim.

- The claim that religion and spirituality should be part of telehealth design (line 693) appears culturally inappropriate without contextual evidence or stakeholder analysis. [ref 13: Malaysian Data]

Conclusion: The paper suffers from advocacy bias—making aspirational claims instead of evidence-based conclusions.

5. Poor Structural Organization and Redundancy

There are multiple redundancies, and the paper lacks tight thematic organization.

- The same benefits and barriers are repeated across multiple sections (e.g., section 3.3 vs. Table 2 vs. section 4.1).

- The distinction between “telehealth,” “telemedicine,” “telerehabilitation,” “TeleGeriatrics,” and “TeleGerontology” is unnecessarily labored, yet inconsistently applied throughout the paper.

- The discussion reads like an informational pamphlet rather than an academic synthesis.

6. Minimal Caribbean Representation

The title promises a review covering “Latin America and the Caribbean,” but the Caribbean is virtually absent. [Mainly mentioned in the titles. No relevant content.]

- Out of 84 documents, almost none focus on Caribbean nations or telehealth experiences specific to their infrastructure, cultural context, or healthcare system constraints.

- The vast majority of examples are from Chile, Mexico, and Brazil.

- This discrepancy between title and content misleads readers and further reduces the value of this review as a regional reference.

Author Response

Dear Reviewer,

Thank you very much for your comments, which allow us to improve our scientific article for publication in this scientific journal.

The attached Word document contains the point-by-point responses to your comments.

Thank you in advance. We look forward to your comments.

Best regards,

Reviewer 2 Report

Comments and Suggestions for Authors

Many thanks for the opportunity to review this interesting and well written article. The results and discussion are interesting and would be informative for larger conversations in the area of telehealth. My main feedback is pivoted on missing data/clarity of methodologies.

MAJOR:

  1. Abstract: there are no methods mentioned. Would recommend re-writing abstract in a structured format to ensure all information is included.
  2. Introduction: I appreciate that the authors spend some time in the results describing the definitions used for telehealth – however would also recommend introducing a telehealth definition in the introduction (there are >100 definitions for telemedicine alone and probably more for telehealth at this stage).
  3. Methods: Create an appendix, and include an example of a search string used within the appendix. Also is there any Boolean logic applied to this search string (AND/OR).
  4. Methods: Please include an “Inclusion and Exclusion Criteria” table to outline how you came to these manuscripts (can also be in appendix if required).
  5. I note that you are following the methods from references 43 + 44 – but can you describe what these are and if there is an associated reporting checklist?
  6. Results: A table is needed to describe the identified manuscripts. This can again go in the appendix if limited by figure/table count. This table should include at a minimum: Title of paper, country of focus, year, the type of publication (abstract, journal, conference, etc), and the type of research (intervention, review, implementation) etc.

Minor:

  1. Were there any bias/quality appraisal tools used for this review? If not, please explain why they were not needed.
  2. Please list or describe what databases/resources were used for the Government documents.
  3. For Figure 2 – is this a design of the authors on creation, or is this a summary of how the found manuscripts describe these terms and relationships?
  4. For the Discussion – the authors mention “Digital Literacy”, but there is more of a trend now to focus on “Digital Health Literacy”, which the authors may wish to include here.

Author Response

(The authors gave the same response as above.)

Reviewer 3 Report

Comments and Suggestions for Authors

Dear Authors,
The paper presents a very comprehensive report of the learnings and observations now available regarding telehealth and the older population from and since the COVID19 Pandemic. The text is extremely detailed, containing a high degree of critical information among a much larger amount of descriptive content. The paper is a valuable and well presented body of work. A pleasure to find so much valuable and transferable observation in one place.

Overall the extent of the content obscures the clarity regarding central trends in the insights, clinical recommendations and remaining challenges. Providing a clearer thread, or differentiation of describing everything at the cost of allowing the critical recommendations to be focused on should be considered.

Please also consider how to address the clinical implications of the recommendations being made in this paper. Should there be consolidation of the recommendations so that a central comment can be made on them, together?

Minor comments:
Line 158 – please explicitly name the sources, or name the type of method as referenced by…
Line 166 – a reference librarian contributed to what degree exactly? Verified all steps? The methodology? The quality or consensus decisions?
Line 179-181 – ‘were identified’?
Line 181-183 – reference 47 says a focus on prevention? The given source is also not global. Consider further references for this point and if they really do focus on prevention
Line 432 – OP users? Or OP telehealth users?

Author Response

(The authors gave the same response as above.)

Round 2

Reviewer 2 Report

Comments and Suggestions for Authors

Dear authors, many thanks for your work on this manuscript. I feel this edits/changes have really strengthened this work and I am happy to accept at this stage.

Reviewer 3 Report

Comments and Suggestions for Authors

Well done on an excellent revision. The paper is much stronger, the value in the content much clearer to find and apply.